# Women’s Health Information-Seeking Experiences and Preferences for Health Communications on FDA-Regulated Products: A Qualitative Study in Urban Area

**DOI:** 10.3390/ijerph21030321

**Published:** 2024-03-09

**Authors:** Moaz Abdelwadoud, Jennifer Huang, Ester Villalonga-Olives, Susan dosReis, Liz Jansky, C. Daniel Mullins, Marc Kusinitz, Heather Ovelmen, Julia Ju

**Affiliations:** 1Department of Global and Environmental Health, New York University, New York, NY 10003, USA; moaz.abdelwadoud@nyu.edu; 2Westat, Rockville, MD 20850, USA; jenniferhuang@westat.com (J.H.); lizjansky@westat.com (L.J.); 3Department of Practice, Sciences, and Health Outcomes Research, University of Maryland School of Pharmacy, Baltimore, MD 21201, USA; ester.villalonga@rx.umaryland.edu (E.V.-O.); sdosreis@rx.umaryland.edu (S.d.); 4United States Food and Drug Administration, Silver Spring, MD 20993, USA; marc.kusinitz@gmail.com (M.K.); julia.ju@umaryland.edu (J.J.); 5National Institutes of Health, Bethesda, MD 20892, USA; heather.ovelmen@nih.gov

**Keywords:** women, aging, health communication, consumer health information, United States Food and Drug Administration, FDA

## Abstract

A key part of any effort to ensure informed health care decision-making among the public is access to reliable and relevant health-related information. We conducted focus groups with women from three generations across the Baltimore–Washington metropolitan area to explore their information-seeking motivations, perceptions, challenges, and preferences regarding three FDA-regulated products: drugs, vaccines, and medical devices. The youngest generation discussed seeking health information for their children; the other two sought information for their own needs. All participants noted that finding health information appropriate to their reading level was a challenge, as was identifying reliable sources of information. All generations identified in-person and live interactions as their preferred method of communication and health care providers as their preferred source for information. All three generations recognized the usefulness of websites, and the two older generations acknowledged the advantages of brochures. Our findings suggest approaches the FDA could consider to improve communications: (a) supporting in-person and live health information interactions; (b) leveraging the agency’s standing with the public to highlight it as a leading source of validated health information; (c) increasing the FDA website’s visibility in internet searches and making its navigation easier; and (d) using multi-pronged approaches and media for various audiences.

## 1. Introduction

In general, women in the United States shoulder much of the responsibility for the health care of themselves and their families [1]. In 2018, 89.3% of American women reported seeking services or advice from a health care facility [2]. About 80.1% of American women aged 55 years or older have one or more chronic conditions [3]. Despite women’s health care needs and the high utilization of health care services, women are also found to delay or not receive health care [4]. Caregiving is an added burden on women. The 2017 Health Information National Trends Survey (HINTS) revealed that 64% of women look for health information to support someone [5].

The unmet health needs and potential caregiving responsibilities among women underscore the importance of having ready access to health information to support the health-related decisions they make for themselves and for persons in their care. Accessing reliable health information, however, can be a major challenge for anyone in our concurrent digital age [6]. Despite the importance of understanding the health information needs of women in older generations, limited research to date addresses their health information-seeking motives, perceptions, challenges, and preferences regarding FDA-regulated products. Understanding these health information-seeking elements is key to FDA’s mission for improving communication strategies and materials in order to help the public, including women, make better-informed health decisions.

This study aimed to (a) identify motives, perceptions, challenges, and preferences among women in older generations for health information sources and materials related to FDA-regulated products, including drugs, vaccines, and medical devices; and (b) explore their preferred information sources and materials, including variations by generation and caregiving status.

## 2. Methods

### 2.1. Study Design

In August and September 2018, we followed a modified grounded theory approach to conduct in-person focus groups among women from the Baltimore–Washington metropolitan area to elicit their motivations, challenges, and preferences toward health information sources and materials associated with FDA-regulated products.

### 2.2. Participants

The study participants included women from three generations: Generation A (born 1965 to 1980), Generation B (born 1946 to 1964), and Generation C (born 1928 to 1945). Study eligibility criteria were (a) self-identified women and (b) women between 38 and 90 years old in 2018. 

### 2.3. Materials and Methods

Our moderators conducted semi-structured focus groups to probe for potential underlying assumptions that could give rise to particular views and opinions. Each focus group had two parts: first, participants spoke of their health information-seeking behaviors related to FDA-regulated products; second, participants shared their thoughts, preferences, and recommendations about three examples of communication methods that the FDA uses to disseminate health information to the public. These communication methods included a brochure describing the Vaccine Adverse Event Reporting System (VAERS), an FDA webpage describing safety communication for Biotin, and an FDA drug safety podcast explaining the adverse effects of a migraine patch. We assessed whether participants’ motivations, preferred information sources, and materials varied by product type. Participants were also given the opportunity to discuss other relevant thoughts or concerns and to provide input about other types of communication not previously discussed. 

### 2.4. Recruitment

Using a convenience sampling strategy, recruitment was conducted by the community engagement team of the Patient-Centered Involvement in Evaluating the Effectiveness of Treatments (PATIENTS) Program at the University of Maryland School of Pharmacy. The PATIENTS Program recruited women through their social networks and community-based organizations. The PATIENTS Program’s community-focused approach engages patients, care providers, and local communities in West Baltimore and beyond, especially those from underserved and minority populations and in patient-centered-outcomes research [7]. 

The community engagement team collaborated with its network of community-based, faith-based, health care, and senior housing facilities across the Baltimore–Washington metropolitan area to recruit participants and host focus groups in locations convenient to participants, with the goal of recruiting six to eight participants for each focus group. Evidence from previous studies indicates that 80% of prevalent themes are discoverable within two to three focus groups, and 90% are discoverable within three to six focus groups [8]. Thus, the goal was to conduct three to four focus groups per generation.

At each collaborating site, we implemented the following recruitment steps: (a) identifying the appropriate age group for each focus group site; (b) identifying convenient focus group venues; (c) scheduling focus groups; (d) tailoring the recruitment flyer for each focus group; (e) reaching out to potential participants via in-person communications, phone calls, and newsletters; (f) screening interested women; and (g) enrolling women who met the eligibility criteria.

### 2.5. Procedure and Data Collection

To ensure anonymity and cultivate the trust needed for an open discussion, we offered participants the option of not using their full names, using an alias, and not disclosing any identifiable health information during the discussions. For confidentiality, we asked participants to refrain from sharing information with anyone outside of the focus group. Each participant was provided a $40 gift card as a token of appreciation for their participation. 

Prior to each focus group discussion, participants completed a brief demographic survey. Three qualitative researchers facilitated the focus group discussions, and one researcher co-facilitated the discussions and took notes. Audio recordings of the focus groups were transcribed meaning to meaning, and all participant records were kept confidential.

### 2.6. Data Analysis

Data collection and analysis occurred concurrently. Thematic data saturation was evaluated throughout this concurrent process by assessing whether new focus groups repeated the topics and themes expressed in other groups of the same generation. Transcripts, and facilitator and co-facilitator notes were imported into NVivo 11^®^ software (QSR International, Burlington, MA, USA) for analysis. Following a stepwise inductive thematic analytic approach, two researchers developed a codebook for analysis, independently coded the focus group data, identified conceptual themes, and discussed discrepancies in coding. Themes were discussed among the research team for overall group consensus.

Results were organized by the five main topics of the focus group guide: (1) motivations and purposes for seeking health information; (2) challenges in seeking health information; (3) preferred methods and sources for health information; (4) preferred communication materials for FDA-regulated products; and (5) suggestions to improve FDA communication materials. Additional topics that emerged during the discussions were also included.

### 2.7. Ethical Approval

The University of Maryland, Baltimore Institutional Review Board (IRB), and the FDA IRB approved the study protocol. Informed consent was obtained from each participant before the beginning of each focus group.

## 3. Results

A total of 109 women participated in 13 focus groups, and each discussion lasted 1.5 to 2 h. One-third or 33% of participants were from Generation A, 27% were from Generation B, and 40% were from Generation C (Table 1). The majority (66%) of participants self-identified as African American, 36% identified as White, 3% preferred not to answer, and less than 2% identified as Native Hawaiian/Pacific Islander or Hispanic/Latino. About a quarter (24%) of participants self-identified as caregivers. 

Results were organized by the main topics of the focus group guide (Appendix A): (a) motivations and purposes for seeking health information; (b) perceptions about health information sources; (c) challenges in seeking health information; (d) preferred methods and sources for health information; and (e) preferred FDA health communication materials. 

Two unsolicited topics emerged from the discussions: (a) impact of religion and spirituality in health decision-making and (b) use of complementary and alternative medicine. These topics were spontaneously discussed by participants in the first focus group with Generation C and were also raised by participants in several of the subsequent focus groups. Thematic saturation was reached as no new themes were identified after the fourth focus group of each generation.

Extracted themes and example quotes are reported below, along with differences found between generations and FDA-regulated products by caregiving status. Participants’ quotes do not necessarily reflect the opinions of the study researchers, FDA, or the United States Government. Table 2, Table 3, Table 4 and Table 5 list and compare subtopics and themes by generation. In these tables, a range of the number of times each theme was endorsed by each generation is presented: none; several (1–10 times); some (11–20 times); and many (21 times or more). Only verbally expressed opinions were reported in the tables. Non-verbal responses (e.g., head nodding) were not captured in the transcripts, and thus not included in these ranges. 

### 3.1. Motivations and Purposes for Seeking Health Information

Participants’ motivations and purposes for seeking health information were identified, differentiating between information for personal use and caregiving (Table 2).

#### 3.1.1. Information for Personal Use

Understanding drug side effects, effectiveness, and interactions was the primary purpose for seeking health information. Participants across all three generations noted that their primary purpose when seeking health information is to understand their medications’ risks, necessities, and interactions. For instance, one participant stated

“I’m allergic to a lot of medication. Even though the doctor can prescribe things, I can’t take anything with morphine, codeine, or any of that in it. So, I have to be my own guardian about that stuff. I have to read those things because I need to know what’s in it!” (Generation C)

Participants expressed their concern about being overprescribed by providers and their need for informational support. One participant noted

“I’ve been on two different kinds of high blood pressure pills for years. I’ve never understood why I’m on two. I would have thought a higher dose of one would make more sense. I just went to the clinic the other day and saw a new doctor and he gave me a third one! The bottle is still sitting there unopened. I’ve been asking people I know in the health industry, and some say I should, and some say I shouldn’t take it.” (Generation A)

#### 3.1.2. Information for Caregiving to Others

Younger women sought information as caregivers regarding their children’s vaccines, medications, and food allergies. As caregivers for young children, several participants from our Generation A and Generation B groups spoke about seeking information pertaining to the human papillomavirus (HPV) vaccine, pediatric medications, and food allergies from multiple sources, including their providers and the internet. Generation C participants did not report health information-seeking for caregiving purposes. A participant from Generation A stated

“When it’s your kid, you do tend to read more about it before you say that, that medicine is OK for them, even if it’s an antibiotic. It’s something that you tend to read more about versus than saying OK, fine”. 

#### 3.1.3. Perceptions about Health Information Sources

Participants discussed their opinions on health information from different sources, including the FDA, pharmaceutical industry, health care providers, and sites on the internet. Trust and trustworthiness were defining features of the participants’ perceptions about these entities (Table 3).

##### Trust and Trustworthiness

The FDA was perceived as a reputable organization and the “FDA-approved” sign was perceived as trustworthy. Across all generations, several participants viewed the FDA as reputable and trusted FDA-approved products. Nevertheless, a few participants who expressed trust in the FDA were not familiar with the exact roles and responsibilities of the agency. Participants noted the following:

“I trust it (FDA) because I’ve seen it all my life. Do I know what the FDA is? No. But I trust it because I’ve seen it on food, medical facilities, all that.” (Generation A)

“There are less pictures (in FDA materials). It just seems more like a required document, like someone put time into it, there’s a format they follow, and standards. It makes me trust it more.” (Generation B)

##### Perceived Conflict of Interest

There was a perceived conflict of interest between the government, pharmaceutical industry, and health care providers. Of the participants indicating that the FDA was a trusted source of information, several from all three generations discussed the FDA’s perceived conflict of interest, which reduced their ability to trust information from these sources. 

Several discussions pointed to a lack of trust in the government in general, questioning the trustworthiness of the FDA as a government agency and its links to the pharmaceutical industry. These participants believe that the FDA had a financial incentive to support and advertise certain pharmaceutical products. In addition, several participants discussed their skepticism of FDA information because of past mistakes (e.g., drug recalls). Other participants said that they do not understand the role of the FDA and the agency’s regulatory procedures are not always transparent. For others, physicians were also viewed as untrustworthy, e.g., paid to recommend specific drugs. These two statements underscore participants’ skepticism:

“When you hear things about drugs or devices on the news, why isn’t the FDA coming to us? Why can’t we hear about it before a mass lawsuit? It would get to consumers quicker so they can make informed decisions before hearing that 20,000 people died.” (Generation A) 

“The pharmaceutical companies and the FDA are all basically one and the same in many respects. The doctors, naturally, listen to the pharmaceutical companies. They owe—the borrower is server to the lender.” (Generation C)

##### Financial Interest

There was a lack of trust in the pharmaceutical industry based on financial interest. Several participants attributed their lack of trust in pharmaceutical companies to their perception that a number of these companies have unethical practices and intentionally raise prices of essential medications and devices.

“A lot of people can’t afford diabetes medication but so many people need it! It seems like there are common illnesses and drug companies will jack the prices up on them.” (Generation C)

##### Time and Transparency

Familiar health care providers were noted as trustworthy, but it took time and transparency to build this trust. Although all generations spoke about generally trusting their health care providers (e.g., primary care physician), they expressed concern that many physicians may not have the most updated information about medications and adverse effects. Participants felt comfortable if they were able to ask their provider questions and build relationships over time. 

“For my personal doctor, I’ve been with her since I was 16. She has shared with me over the years what she does to keep herself apprised of the new information. I trust her now, but that’s a long-time relationship.” (Generation A)

“I question every doctor. If you get an attitude or upset because I’m asking you a question about your profession, we’re done. Even when you go to the pharmacy, you have to know your health.” (Generation A)

##### Verification of Internet Sources

Generations A and C were skeptical about most health information they found on the internet and said that it should be considered with caution and verified by “reputable” sources. 

“I don’t always agree or trust what I get on the internet. I do some examinations for myself and then make decisions.” (Generation C)

### 3.2. Challenges in Seeking Health Information

Several health information-seeking challenges were discussed in our focus groups. We categorized these challenges and examples under two main headings: (a) comprehension and (b) sufficiency (Table 4). 

#### 3.2.1. Comprehension of Health Information

Medical information was not written or formatted appropriately to be comprehended well by all patients. Participants from all generations very often agreed that health communication materials, particularly medication package inserts, are often not appropriate for all patients because the information is presented using technical language and a small-sized font.

#### 3.2.2. Sufficiency of Health Information

Overwhelming information was a barrier to finding specific health information. All three generations spoke about the difficulty of obtaining sufficient health-related information from one source. Participants often sought information from many sources, including different health care providers, internet sites, peers, and family members. Generations A and B expressed that many sources could be helpful. However, making sense of this voluminous information was overwhelming, as was the need for thorough validation to ensure that all the information was reliable and useful. For example, participants noted the following:

“… I need to talk to every one of my doctors and figure out if the dose on mine is still good. Then I have to put all their information together because they won’t all agree. They won’t all say the same thing. I’ll come back around and ask more questions. A year later, I might have my answer or what I’m comfortable with.” (Generation A)

“I find there’s an overload of information, not that there’s a lack of. You’re going to get 500 websites talking about whatever subject you put in. Then you have to filter through that to try to get the information that you want.” (Generation B)

### 3.3. Preferred Methods and Sources for Health Information

Although our participants used the term “sources” to refer to the sources and methods of health information delivery, we differentiated them in our thematic analysis (Table 5). 

#### 3.3.1. Preferred Methods

##### In-Person and Live Interaction

In-person and live interaction was the preferred method to receive health information. Participants from all generations identified in-person (e.g., face-to-face) and live (e.g., phone calls, telemedicine) interactions as the best method for obtaining answers to health-related questions. Examples of useful in-person and live interactions included speaking with health care professionals over the phone to answer specific health questions, asking pharmacists for details about drugs, and discussing diseases and health conditions with physicians. For instance, participants stated the following:

“The nurses can usually explain to you what you’re taking the medicines for or if you have any other kind of issues. I would suggest they do the nurse line rather than the website.” (Generation B)

“If they have available staff there to answer the question, then this would be a good thing. Some people do better talking with somebody on the phone than reading.” (Generation B)

#### 3.3.2. Preferred Sources

##### Personal Health Care Providers

Overall, health care providers were the main and preferred source of health information on the three types of FDA-regulated products for all generations. Participants reflected on their personal interactions with their health care providers and said that the health information they receive through these interactions is the most useful and trustworthy. Although the FDA does not regulate the practice of medicine, some participants were concerned about their providers’ limited time and knowledge about their health concerns and the lack of communication between primary care and specialized providers. One participant said

“Doctors don’t tell you everything. You’re in there for a 15-min office visit, you forget the question you wanted to ask, so you get home, look up everything you want to know, then when you go back to the doctor, you can go over it.” (Generation C)

#### 3.3.3. Utilized Sources

##### Internet

Participants from all three generations reported frequent use of the internet to find health information. Some women in Generation A and a few in Generations B and C cited different purposes for using the internet, including confirming providers’ information, preparing for medical appointments, and finding general information about their symptoms. Many participants who spoke about internet use stated that a “Google search” was their gateway for internet searches. The following websites were mentioned by participants: WebMD, Mayo Clinic, health insurance companies, support groups, MedlinePlus, Dr. Weil, FDA, pharmaceutical companies, National Center for Homeopathy, ABC Homeopathy, YouTube, and Facebook. Typical responses were as follows:

“First, I go to my internist or other specialty doctor, then I reinterpret what they tell me through Google.” (Generation C)

“You can research your symptoms, see what type of medication they may give you, then you go to the doctor and you’re ready to hear the options. You already have some information you’ve collected for yourself. That’s how I prepare myself.” (Generation A)

##### Social Media

The youngest generation was most likely to use social media, particularly Facebook, to solicit medical advice or information from family members and peers. However, they said they were cautious when using information from Facebook to guide important health decisions. One participant noted

“Another resource is Facebook. I don’t put a lot of private stuff on Facebook, but I’d ask if anybody knows anything about this.” (Generation A)

##### Health Fairs, Workshops, and Health Expos

Older generations received health information from health fairs, workshops, and health expos. Several participants from Generations B and C reported frequent participation in in-person educational venues (e.g., health fairs, workshops, and health expos). For instance, two participants explained as follows:

“At the health fairs, they take time to explain it to you and answer questions as best they can.” (Generation B)

“[The expo] is once a year, and they give you lots of information about shots and things going on to keep seniors healthy.” (Generation C)

##### Newsletter

The oldest generation was more likely to mention online and printed newsletters as their sources of information. Subscription newsletters included those from the National Institutes of Health (NIH), Cleveland and Mayo Clinics, Brigham and Women’s Hospital, UnitedHealthcare, Seniors Digest, Nutrition Today, and Bottom Line. One woman noted

“I get a lot of information from the Women’s Hospital in Boston. They have a very good newsletter that comes out.” (Generation C)

##### Family Members and Friends

All generations mentioned asking health questions and sharing concerns with family members who work in the medical field (e.g., nurses and physicians). Generation C women spoke about asking younger family members to confirm via the internet the information they received from their physicians, particularly if they are not well-versed in technology and conducting online searches. For instance, participants stated the following:

“My mom is a retired RN, so everybody in the family just goes to her with questions about health.” (Generation A)

“I go to the doctor a lot. I ask questions even when I don’t understand. I ask them to explain it to me. I write it down. I take it to my daughter who helps explain it to me.” (Generation C)

### 3.4. Preferred Communication Materials for FDA-Regulated Products

In the second part of each focus group, we presented three examples of communications (website, brochure, and podcast) that the FDA uses to disseminate health information. Presenting these examples to participants was intended to generate discussion and elicit thoughts and recommendations on preferred communication platforms. The feedback on these materials is as follows.

#### 3.4.1. Websites

Overall, participants from all generations found websites to be useful because they were comprehensive and available to all types of learners. One participant said

“A lot of people love the internet. They are quick on it. The world is in your hands, right here in this phone.” (Generation A)

Some participants noted that websites were particularly important when seeking information about prescribed drugs and their contraindications. A typical response was

“Websites would be a good place to go to learn about side effects, dosages, what causes medicine interactions.” (Generation B)

#### 3.4.2. Brochures

Brochures were useful for older generations. Generations B and C discussed the benefits of brochures, including being available to reference them later, sharable with other people, a summary of information, a reference for finding more details, a physical/visual reminder from providers, and a tool for generating conversations with providers during health care visits. For instance, one participant noted

“You know, [the doctor] can’t tell you everything in just 15 min so I think it’s a good thing. When you get home, you can read over something at your own pace.” (Generation C)

Generation A expressed concerns about brochures, including advertising that makes content about vaccines or drugs untrustworthy, overuse of pictures that impact legitimacy, and wasting environmental resources. One woman noted

“If the pictures were real, that would help. In 20–30 years, we might trust pamphlets more. Then paper will be obsolete, so there’s no point.” (Generation A)

#### 3.4.3. Podcasts

Podcasts were deemed the best for announcements, multitasking, and tech-savvy users. Although participants were skeptical about using podcasts, all generations referred to podcasts as a good source for disseminating public announcements and notifications (e.g., drug recalls) to aural learners and to those in multitasking situations. Older generations mentioned they would recommend podcasts to younger audiences who were technologically savvy. As one participant explained,

“We can simultaneously do three or four things … You can pick it up through your multitasking … when you hear something that gets your attention, if you can go to it right then, you will.” (Generation B)

### 3.5. Suggestions to Improve FDA Communication Materials

#### 3.5.1. Multiple Materials and Approaches

Participants suggested that the FDA should employ a variety of communication modes, formats, and approaches given the variability in individual learning styles, skills, access, and preferences. One participant noted

“I don’t think it’s all the same. I think diversity is what we need … We need to have different things. This might work... You might want to look at all three [methods] and draw something from each one.” (Generation B)

#### 3.5.2. Website Enhancements

Suggestions for improving the utility and accessibility of the FDA website included ensuring that the website appears at the top of Google searches for health topics, that the FDA website’s search engine is optimized, and that information is organized with drop-down menus. For example, participants said the following:

“I didn’t realize till I heard about this project that there was a website [for FDA]. When I looked, I couldn’t believe how much information was there!” (Generation C)

“You need to know what you’re looking for when you go to the FDA site. If you don’t, it’s overwhelming.” (Generation A)

### 3.6. Emergent Themes

Two topics relevant to health information-seeking and health decision-making emerged during our focus groups without prompting from the moderators. We organized those discussions under two subtopics: (a) religion and spirituality in relation to health decision-making and (b) complementary and alternative medicine. 

#### 3.6.1. Religion and Spirituality

Health decisions were often guided by religious beliefs and spirituality. Some participants across all generations cited their faith in God to take care of their health and reported that their religious practices and beliefs guide their health care decisions. Some participants in Generations A and C said their spirituality plays an important role in their health-related decision-making and that they engage in spiritual activities like meditation and yoga to improve their ability to handle their medical decisions. For example, one participant commented

“I know the doctors are His helpers. That’s how I look at it. He’s got the first and last word when it comes to those decisions.” (Generation A)

#### 3.6.2. Complementary and Alternative Medicine

Participants had a preference for complementary and alternative medicine. Many participants from all generations said they or their family members use alternative medicine or remedies from other countries in addition to, or in place of, Western medicine. In the Generation C groups, some participants discussed the efficacy of these treatments and FDA’s role in their approval and regulation. Participants said the following:

“I’ve been using holistic and natural remedies since 1974. Yes, sometimes we need allopathic and there are some really good doctors, but the FDA puts a lot of fear out there in order to keep the pharmaceutical companies going.” (Generation C)

“My parents are 74 years old. They don’t look like it though. They’re more home remedy people. They don’t like the hospital and don’t want to go there. They have a home remedy book and they’ve passed it around my family. Some of these things really do work.” (Generation A)

## 4. Discussion

Understanding women’s motivations for seeking information, perceptions of the validity and usefulness of information sources, challenges in finding and understanding information, and preferred health information sources may help to improve the FDA’s regulated product health communications for women [9,10]. According to the findings from this study, the specific information that women seek and how they wish to receive it depended on their motives for seeking the information, as well as their individual preferences. The primary reason that women sought health information was to locate reliable data about prescribed medications. The primary reason caregivers of children sought health information was to find reliable data about children’s vaccines, medications, and food allergies. 

How women seek health information varied across individuals, in part because of their ages [10]. Our findings suggested there are generational differences in the most used information sources and resources. For instance, younger women preferred internet sources and older women preferred in-person educational venues, friends, and family members. 

The following key findings are based on our analysis of the focus group interviews. First, trust in health information sources was an important topic in our discussions. While several participants perceived the FDA as a reputable organization, a lack of trust in the government and the agency’s perceived conflict of interest with pharmaceutical companies jeopardize this positive image. Nonetheless, participants’ greater trust in the FDA compared to the federal government resembles the findings of Kowitt et al. (2017), which showed that 62.5% of adult Americans trusted the FDA, while only 42.9% trusted the federal government [11]. Similar findings showed that less than 20% of Americans reported that they trusted the federal government, compared to the 70% that reported that they viewed the Centers for Disease Control and Prevention (CDC) favorably [12]. In addition, evidence indicates that official governmental bodies are effective in fighting health misconceptions via providing corrective information [13]. In addition, previous research suggests that the authority of the health information owner has a positive impact on both its credibility and trust [14]. Collectively, this suggests that public trust in the FDA presents an opportunity for the agency to leverage its reputation as a leading source of reliable and validated health information. A second important topic was information sources [15,16]. All generations identified in-person and live interaction as their most preferred method and their health care providers as their most preferred source for health information. All other sources, including the internet, social media, newsletters, family members, and friends, came next in preference. In addition, participants reported facing challenges in finding and understanding health-related information from various sources. For example, with the increasing access to and use of the internet, participants spoke of being overwhelmed with health information and having to discern useful and truthful information. Others reported receiving conflicting information from their primary care provider and specialists, contributing to frustration and confusion. 

Additionally, many participants in Generation A, some in Generation B, and several in Generation C indicated that the FDA must consider appropriate reading levels to address health literacy needs for all patients. Another approach supported by this study is coordinating with health care providers, faith communities, and community health fairs to facilitate patients seeking health information from the FDA via in-person or live interactions.

A key finding of the focus group discussions was the general suggestion for the FDA to consider diversifying its communication materials to reflect the variety of sources and materials that different generations use. Suggestions for improving FDA’s website underscore the need to improve the layout and features of health information websites, which would make it easier for diverse populations to understand [17]. Improving the website readability and simplifying its content would improve the perception of the information without impacting the trust in the source [14]. One way to improve access to the FDA’s health communications is to make its website more prominent on internet search engines [18,19].

Each generational group in our study used available health information sources in its own way. Older generations reported greater use of in-person educational venues (e.g., health fairs and newsletters). The youngest generation reported more frequent use of the internet, including social media, compared to older generations. These findings are consistent with data from the Women’s Health Initiative cohort study indicating that only 60% of women aged 65 and older used the internet as a source of health information, and that women who used the internet as an information source were more likely to be younger, be non-Hispanic White, earn a higher income, have a higher educational level, and live with a partner [20].

Specifically, participants reported receiving health information and advice from their friends and peers, partially through social media. These findings echo those from the HINTS 2013–2017 analysis, which showed that younger populations were more likely to use social media for health communication [21] and that women tended to share health information on social media and online support groups for people with similar health conditions [5]. 

The impact of religion and spirituality on participants’ health decisions were not surprising. Previous research found that religious beliefs can guide decision-making among older adults and represent a major coping aid during and after medical decision-making for critically ill patients [22,23]. Similarly, the preference for complementary and alternative medicine aligns with the National Health Interview Surveys’ results that showed increases in the use of yoga, meditation, and chiropractic therapy from 2012 to 2017 among American adults [4]. 

### Limitations

As with all focus group studies, results from our study should not be generalized to the broader population and should be comprehended within our participants’ demographics [24,25]. Another limitation of our study is its lack of demographic and geographic diversity; more than 60% of participants were African Americans, and all participants were from the Baltimore–Washington area. Another limitation was that due to time constraints, we focused discussions only on caregiving for children and not on caregiving for spouses, parents, or older family members. 

The examples of FDA health communication materials (brochure, webpage, and podcast) were provided to showcase different health communication methods and may not be representative of all FDA health communications. Note that the content is different in each example. Some participants likely were not familiar with specific products in the examples, whereas those who go on the FDA website might seek information on products for their conditions. It is possible that the product, subject matter, or content of the communication method had an impact on participant preference. Therefore, feedback collected from participants on the examples may not be generalizable to all FDA communications, particularly as the FDA website is dynamic and the agency continues to enhance their public-facing communications.

Finally, these data were collected before the COVID-19 outbreak. FDA has been a visible and important federal agency during the COVID-19 pandemic, especially on the topic of vaccines. Consequently, the use of the FDA website might have changed during the COVID-19 pandemic. Addressing these limitations may help to advance future research in this area.

## 5. Conclusions

Overall, our findings suggest broadly the ways that the FDA can improve regulated medical product health communications intended for women. This focus group study indicated four specific options to consider. First, public trust in the FDA logo suggests that the agency could leverage its image as a leading source of reliable and validated health information. Second, in-person and live interactions were found to be the preferred method to receive health information. The FDA could explore ways to support these communications which may also enhance the image of the agency’s direct communications with patients and caregivers. Third, the FDA could investigate ways to be more visible on major internet search engines and identify ways to improve their website navigation. Lastly, the FDA could offer different media and communication strategies for conveying health information [26] that accommodate different preferences for a variety of women, including images of diverse women in terms of race/ethnicity and age. 

Our findings should inform future quantitative research, such as a national survey, aimed at collecting nationally representative data to identify new strategies for improving health communications designed for women in the United States and to further investigate the nuances among preferences for health communications for the various types of FDA-regulated products. 

## Figures and Tables

**Table 1 ijerph-21-00321-t001:** Demographic characteristics of focus-group participants (*n* = 109).

Demographic Characteristics	Number of Participants	Percentage of Participants
** *Generation* **		
Generation A: Born 1965 to 1980; 38 to 53 years old in 2018	36 *(5 focus groups)*	33.0%
Generation B: Born 1946 to 1964; 54 to 72 years old in 2018	29 *(4 focus groups)*	26.6%
Generation C: Born 1928 to 1945; 73 to 90 years old in 2018	44 *(4 focus groups)*	40.4%
** *Race* **		
African American	66	60.6%
Native Hawaiian/Pacific Islander	2	1.8%
Hispanic/Latino/Spanish Background	2	1.8%
White/Caucasian	36	33.0%
Did not prefer to answer	3	2.8%
** *Education* **		
Less than high school	15	14.0%
High school	35	32.7%
Some college education	18	16.8%
College degree	28	26.2%
Postgraduate education	11	10.3%
Did not prefer to answer	2	1.9%

**Table 2 ijerph-21-00321-t002:** Motivations and purposes for seeking health information: subtopics and themes by age group.

Subtopics	Themes	Generation A *	Generation B	Generation C
**Information for Personal Use**	Information about drugs’ side effects, effectiveness, and drug interactions is the primary purpose for seeking health information	Several **	Several	Several
**Information for Caregiving to Others**	Younger women seek information as caregivers for their children’s vaccines, medications, and food allergies	Several	Several	None

* Generation A: 38 to 53 years; Generation B: 54 to 72 years; Generation C: 73 to 90 years; ** Number of times the theme was endorsed by this generation: None = 0, Several = 1–10, Some = 11–20, and Many = 21 or more.

**Table 3 ijerph-21-00321-t003:** Perceptions about health information sources: subtopics and themes by age group.

Subtopics	Themes	Generation A *	Generation B	Generation C
**Trust and Trustworthiness**	There is a perceived conflict of interest between the government, the pharmaceutical industry, and health care providers	Several	Several	Several
	The FDA is a reputable organization and the “FDA approved” logo is trustworthy	Several **	Several	Several
	There is lack of trust in the pharmaceutical industry due to its financial interest	Several	None	Several
	Familiar health care providers are trustworthy, but it may take time and transparency to build this trust	Several	Several	Several
	Internet sources need verification	Several	None	Several

* Generation A: 38 to 53 years; Generation B: 54 to 72 years; Generation C: 73 to 90 years; ** Number of times the theme was endorsed by this generation: None = 0, Several = 1–10, Some = 11–20, and Many = 21 or more.

**Table 4 ijerph-21-00321-t004:** Challenges in seeking health information: subtopics and themes by age group.

Subtopics	Themes	Generation A *	Generation B	Generation C
**Comprehension of Health Information**	Reading level of medical information is not appropriate for all patients	Many **	Some	Several
**Sufficient Health Information**	Overwhelming information is a barrier to finding specific health information	Several	Several	None

* Generation A: 38 to 53 years; Generation B: 54 to 72 years; Generation C: 73 to 90 years; ** Number of times the theme was endorsed by this generation: None = 0, Several = 1–10, Some = 11–20, and Many = 21 or more.

**Table 5 ijerph-21-00321-t005:** Preferred methods and sources for health information: subtopics and themes by age.

Subtopics	Themes	Generation A *	Generation B	Generation C
**Preferred Method**	In-person and live interaction is the preferred method to receive health information	Several **	Several	Several
**Preferred Source**	One’s health care provider is the main and most preferred source for health information	Some	Some	Many
**Utilized Sources**	The internet is used frequently for different purposes when seeking health information	Many	Some	Some
	Social media is a supplemental health information source for the youngest generation	Several	None	None
	Older generations receive health information from health fairs, workshops, and health expos	None	Several	Several
	The oldest generation uses newsletters as a source for health information	None	Several	Several
	Discussing health concerns with family members and friends	Many	Several	Several

* Generation A: 38 to 53 years; Generation B: 54 to 72 years; Generation C: 73 to 90 years; ** Number of times the theme was endorsed by this generation: None = 0, Several = 1–10, Some = 11–20, and Many = 21 or more.

## Data Availability

The study code book and redacted data are available upon reasonable request.

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
