# Peer review of "Women’s Health Information-Seeking Experiences and Preferences for Health Communications on FDA-Regulated Products: A Qualitative Study in Urban Area"

_ijerph, 2024, doi:10.3390/ijerph21030321_

Round 1

Reviewer 1 Report

Comments and Suggestions for Authors

My main concern about this paper is that the conclusions (which are presented as generally applicable) overreach the data because of the sample on which the conclusions are based.

As the authors themselves note in the section "Limitations": "Another limitation of our study is its lack of demographic and geographic diversity: more than 60% of participants were African Americans, and all participants were from the Baltimore-Washington area."     Further, I wonder about the sample size.  The overall population of the Baltimore-Washington metropolitan area is 9,764,315 (University of Michigan statistics), but only 109 women participated in the study.

The study is not without value, but should be recast and retitled as a "Qualitative Study in a Large Urban Area."  Further, from a methodological point of view, the authors need to demonstrate that the women who participated are indeed representative of the urban area in question.

Reviewer 2 Report

Comments and Suggestions for Authors

Title

Try to make the title more accurate and shorter

The introduction is too short and does not provide relevent presentation of the problems and the similar studies conducted.

Analysis 

The qualitative analysis is not strong enough to support the questions and problems presented in the introduction. 

Also, we believe some sources should be merged (social media, Internet and websites for example). You first mention internet, then social media then go back to websites. The themes categorization could be further improved. 

Additionally, while important insights could be extracted out of the data, the way it the analysis is done now is very shallow making the study look less strong than it could have been. 

Some titles also are not relevant of the content.

You for instance talk aobut suggestions for improvement in 3.5. and then as subtitles you have multiple materials and approaches, website enhancement, and emergent themes. How is emergent themes part of the improvement suggestions?

Comments on the Quality of English Language

Minor changes are needed

Reviewer 3 Report

Comments and Suggestions for Authors

REVIEW COMMENTS

I congratulate the authors for the topic and the context in which they carry out their research, which is undoubtedly very relevant.

However, further revision is necessary to evaluate the manuscript. The Methodology is not well developed and the error creeps into the Results, Discussion and Conclusions.

I'm going to comment on my revision suggestions.

METHODOLOGY

Study design

It is necessary to incorporate the Theoretical Framework from which the study is approached.

Participants

Define the segmentation criteria for each group of women

Material and methods

This section requires an important modification.

It is necessary to provide detailed information about:

- Focus Group Script

- Criteria used to quantify the number of Focus Groups, why 13?

Incorporate the thematic axes of the discourse/content analysis

How was the information coded?

RESULTS

In Table 1, it is necessary to incorporate the variables Race and Education in a transversal way, they must be incorporated into the horizontal axis to have detailed information about the sample. The way it has been presented is not the correct way to do it.

The tables do not provide detailed information.

Verbatim do not identify with the Focus Group to which they belong

DISCUSSION

As the results are presented, it is not possible to evaluate the discussion.

CONCLUSIONS

cannot be valued

REFERENCES

I suggest incorporating a greater number of references, it is very limited - there are only 22 -

Round 2

Reviewer 1 Report

Comments and Suggestions for Authors

Appropriate revisions have been made in response to my previous voiced concerns.

Author Response

Thank you for the positive feedback and for confirming that "Appropriate revisions have been made in response to my previous voiced concerns."

Reviewer 2 Report

Comments and Suggestions for Authors

Comments addressed.

Comments on the Quality of English Language

Minor revision needed

Author Response

Thank you for the positive feedback. We have proofread the current version of the manuscript to assure appropriate English language. If Reviewer 2 has any specific language or phrase concerns, we would be happy to address those.

Reviewer 3 Report

Comments and Suggestions for Authors

I appreciate the authors' review work, it has greatly improved the manuscript.

However, I see that my suggestions for improvement related to Table 1 have still not been incorporated. As I told you in the first revision: In Table 1, it is necessary to incorporate the variables Race and Education in a transversal way, they must be incorporated into the horizontal axis to have detailed information about the sample. The way it has been presented is not the correct way to do it.

Author Response

Thank you for the positive feedback!

Regarding the request to incorporate into the horizontal axis to have detailed information about the sample, unfortunately this is not allowed by the US Food and Drug Administration. Our FDA co-authors reference the Privacy Act - please see attached document, specifically page 390 where it states the need for "assurance that the record will be used solely as a statistical research or reporting record, and the record is to be transferred in a form that is not individually identifiable;" the interpretation of this is that small sample sizes cannot be reported in cases in which individuals participated in research. This is because additional information could be statistically inferred.

While we recognize that readers may have the same feeling as Reviewer 3, we hope that Reviewer 3 and the editor agree that our manuscript still provides useful information without the requested detailed information regarding participants.
